# Understanding the synergistic effect of physicochemical properties of nanoparticles and their cellular entry pathways

Jiaqi Lin[1,2], Lei Miao[2], Grace Zhong [2], Chih-Hsin Lin[2], Roozbeh Dargazangy[3] & Alfredo Alexander-Katz[1✉]

Gaining precise control over the cellular entry pathway of nanomaterials is key in achieving cytosolic delivery, accessing subcellular environments, and regulating toxicity. However, this precise control requires a fundamental understanding of the behavior of nanomaterials at the bio-nano interface. Herein, we report a computational study investigating the synergistic effect of several key physicochemical properties of nanomaterials on their cellular entry pathways. By examining interactions between monolayer-protected nanoparticles and model cell membranes in a three-dimensional parameter space of size, surface charge/pKa, and ligand chemistry, we observed four different types of nanoparticle translocation for cellular entry which are: outer wrapping, free translocation, inner attach, and embedment. Nano-particle size, surface charge/pKa, and ligand chemistry each play a unique role in determining the outcome of translocation. Specifically, membrane local curvature induced by nano-particles upon contact is critical for initiating the translocation process. A generalized paradigm is proposed to describe the fundamental mechanisms underlying the bio-nano interface.

[1] Department of Materials Science and Engineering, Massachusetts Institute of Technology, Cambridge, MA 02139, USA. [2] Department of Chemical Engineering, David H. Koch Institute for Integrative Cancer Research, Massachusetts Institute of Technology, Cambridge, MA 02139, USA. [3] College of Engineering, Michigan State University, East Lansing, MI 48824, USA. ✉email: aalexand@mit.edu

Nanomaterials' application in biology has ushered in a new era and is changing the way we interface with living organisms. However, the clinical efficacy of nanoparticle has raised concerns, bringing about uncertainty with regards to the superiority of nanomaterials over conventional methods of delivery[1]. Successful delivery of nanoparticle requires overcoming multiple biological barriers to reach the site of action. The persisting inability to surmount two main hurdles that nanoparticles face inside the body – targeting and delivery – largely hinders advances in the therapeutic efficacy of nanomaterials.

In order to surmount these hurdles, it is essential to develop a fundamental, systems-level understanding of interactions at the bio-nano interfaces. Overcoming membrane barriers (including cellular membranes, endosomal membranes, nucleic membranes, and other subcellular organelle membranes) is essential for cellular applications of nanoparticles. For instance, successful cytosolic delivery is a prerequisite for effective probing of the intracellular environment, modulation of gene expression (e.g. using gene therapeutics), and cell-based therapies[2]. Unable to overcome the bottleneck of bypassing cell membranes, emerging therapeutic biologics are restricted to acting on the cell surface, which greatly limits their effectiveness. The success of gene and RNA-based therapeutics also largely depends on the development of safe and efficient cytosolic delivery systems[3]. Furthermore, there is a compelling need to fundamentally understand the interactions at the nanomaterial-cell membrane interface to decipher the cellular entry pathways of nanomedicines which determines their intracellular trafficking cascade and cellular fate.

Extensive studies have been conducted in the past to understand the nanomaterials-membrane interface (e.g studies of nanoparticle size[4,5], shape[6,7], surface charge[8], pKa, and surface chemistry[9,10]). These studies, however, often focus on a single variable[11,12], while in reality, the complex synthetic-biological interface comprises a wide spectrum of biophysicochemical interactions. Therefore, the generalization of the conclusions of these studies to broader situations is difficult. In addition, experimental studies may have large variability in setup and the results often conflict across studies. The scattered results form a quagmire in the advance towards clinical applications. A recent perspective calls for minimum information in bio-nano literature

to reduce variability and to increase quantitative comparison[13]. A systems-level understanding of the interactions that elucidates the collective effect of multiple key physicochemical parameters at the bio-nano interface is urgently needed. To achieve such understanding, techniques and tools that allow quantitative analysis of these parameters also need to be developed[1].

In this manuscript, we aim to understand the nanomaterials-cell membrane interface using a systems approach. Previously we have identified pore-assisted translocation of nanoparticles across cell membranes using computer simulations[14]. Using a larger system and extensive computational time, we here study the synergized effect of size, surface charge/pKa, and ligand hydrophobicity of monolayer-protected nanoparticles on their interaction with cellular membranes. By probing the interactions in this three-dimensional parameter space, we identified four types of translocation of nanoparticles across model cell membranes, featuring distinct cellular entry pathways. In addition, we found local membrane curvature is key in pore formation that permits translocation. We discuss how each of the nanoparticle properties individually and collectively affects its translocation behavior. A generalized paradigm of the nanoparticle-membrane interface is proposed to guide the design of nanocarriers for controlled cellular entry pathway. Lastly, we discuss the distinct intracellular trafficking routes and cellular fates following the observed entry pathways, and the utilization of these subcellular routes in varying cellular applications.

## Results

**Nanoparticle key physicochemical properties**. The nanoparticle-cell membrane interface is a complex interface that involves many biophysicochemical interactions, examples being steric interactions, electrostatic interactions, hydrophobic interactions, solvent interactions, and biological interactions[15].

Key players that affect these interactions and shape the interface include size, surface charge/pKa, and ligand chemistry (Fig. 1). In this study, we focused on nanoparticles with size ranging from 3 nm to 15 nm (Supplementary Fig. 1). Smaller size objects (<1 nm) such as ions and water molecules enter cell membranes via permeation[16]. Larger size objects enter cell typically through endocytosis, which includes phagocytosis, pinocytosis, receptor-mediated endocytosis, and other non-

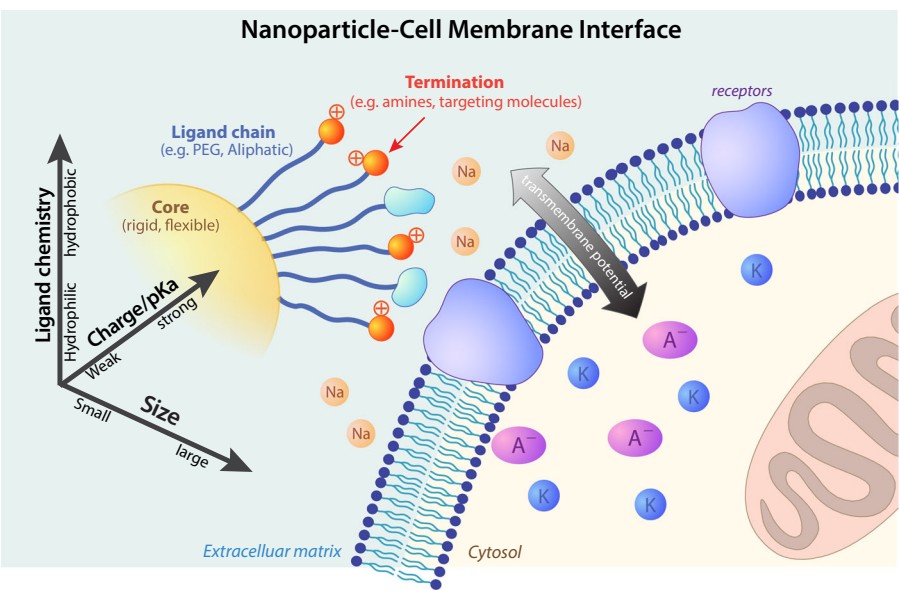

**Fig. 1 The nanoparticles-cell membrane interface.** Size, surface charge/pKa, and ligand chemistry are key physicochemical properties of nanoparticles that shape their interactions with cellular membranes and thus determine their cellular entry pathways.

specific endocytosis[17]. In the middle range that is comparable to the thickness of the cell membrane lies the boundary between permeation (direction translocation) and endocytosis. Protein is in such a size range (from 3 nm to 20 nm) and demonstrates complex interactions with cell membranes[18,19]. Nanoparticles within such size range (~10 nm) show interesting interactions with cell membranes[20]. To study the effect of surface charge/pKa, we use total surface charge as a quantitative measure, as the surface charge of a nanoparticle is the sum of the charge generated from ionizable groups (typically amine) which are determined by its surface pKa and the acidity of the environments[21,22]. The charge is implemented by randomly assigning ligand terminals with an ionized amine group, which emulates ionization of nanoparticles at different pH levels (we assume a neutral pH here)[23]. The maximum ionizable charge on the nanoparticle is capped by the coating of the surface ligand. To study the effect of ligand chemistry, two major ligand types – hydrophobic (e.g. alkyls) and hydrophilic (e.g polyethylene glycol (PEG)) – are considered. We thus have a three-dimensional parameter space of nanoparticle physicochemical properties, resulting in 36 different permutations (Fig. 2a).

The model cell membrane is composed of zwitterionic dipalmitoylphosphatidylcholine (DPPC), which represents the phospholipids in living cell membranes and had been employed in previous simulation studies[24–26]. The transmembrane potential, which plays a key role in interacting with charged and ionized nanoparticles[14], is implemented by adding an ionic imbalance across a double membrane system[27]. The nanoparticle is added above the surface of the membrane. The nanoparticle's position is restrained for 0.2 μs for equilibration before it is released to freely interact with the membrane for 1.2 μs (Fig. 2b).

**Different types of translocation**. We discovered four outcome categories for the interaction between nanoparticles and lipid membranes in the three-dimensional parameter space of key properties investigated. These types of translocation are outer wrap, free translocate, inner attach, and embedment (Figs. 3 and 4). In the category of outer wrap, nanoparticles are wrapped around by the membrane surface to a certain degree but are not able to translocate across the membrane. Outer wrap potentially triggers non-specific or receptor-mediated endocytosis which happens on a longer time scale. This translocation type occurs for nanoparticles with large size and low surface charge (Figs. 3 and 4). In free translocate, nanoparticles completely translocate across the membrane through a pore and enter the cytosol region. Once inside, nanoparticle roams freely and does not further interact with the membrane. Free translocation occurs for nanoparticles with smaller size and higher charge/ionization (Figs. 3 and 4). In embedment, nanoparticles partially translocate and stay embedded in the membrane. Nanoparticles are exposed to both extracellular fluid and cytosol. Such a configuration is similar to that of a transmembrane protein (Fig. 3, lower right) and has been reported previously for anionic nanoparticles coated with alkyl ligands[5]. In inner attach, nanoparticles achieve a high degree of translocation, with most of the nanoparticle surface exposed to the cytosol, but attach to the inner surface of the membrane through a small fraction of the nanoparticle that remains buried inside the membrane (Fig. 3). This type of translocation has not been reported previously. The inner attach configuration only occurs for nanoparticles with hydrophobic ligands and size-charge combinations between free translocate and embedment in the two-dimensional parameter space.

Next, we analyzed the effect of size, surface charge/pKa, and ligand chemistry on the translocation behavior of nanoparticles.

Effect of size: Increasing size generally increases the physical difficulties for the nanoparticle to translocate across the lipid membranes. For instance, nanoparticles with hydrophobic ligand and 100 $e$ surface charge/ionization, as nanoparticle size increases, the translocation type changes from free translocate to inner attach and then to embedment (Fig. 5). Similarly, increasing the size of hydrophilic nanoparticle (with 100 e surface charge/ionization), change translocation type from free translocate to outer wrap. Once the nanoparticle is in the realm of outer Wrap (which leads to endocytosis), size also has an effect. For receptor-mediated endocytosis, nanoparticle size and the density of receptor together determine the rate of uptake[28,29].

Effect of surface charge/pKa: Increasing total surface charge/ionization enhances the driving force, which renders the nanoparticle more likely to translocate. Our simulations showed that as surface charge/ionization increases, the translocation type changes from outer wrap to embedment, then to inner attach, and

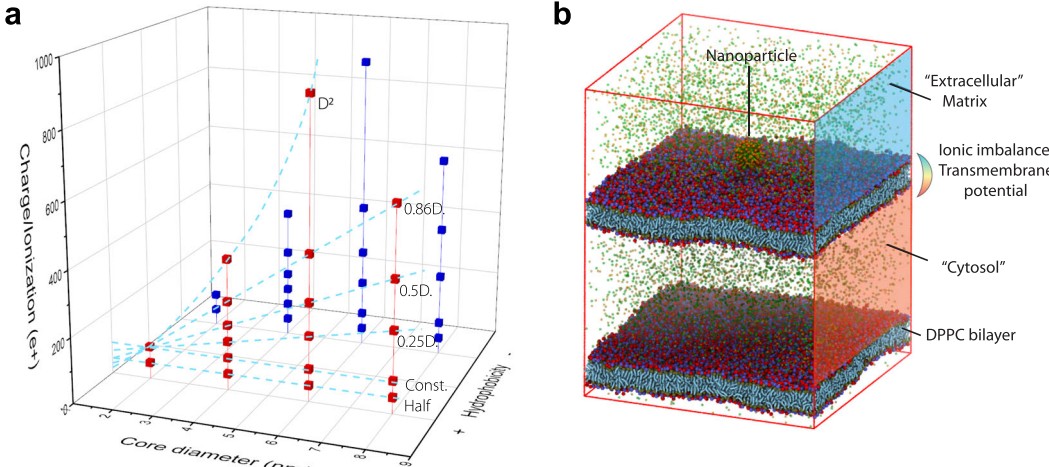

**Fig. 2 Nanoparticles with three varying key parameters (size, charge/pKa, and ligand hydrophobicity) and the simulation system including model cell membranes. a** Varying charge-size scaling schemes, including Const, Half, 0.25D, 0.5D, 0.86D, and D² are studied as the nanoparticles increase their core size from 2 nm to 8 nm. The 2-nm core nanoparticle with 99 e ionization is used as a reference. The identical charge-size scheme is studied for both nanoparticles with hydrophobic ligands (red) and hydrophilic ligands (blue). **b** The nanoparticle-membrane systems. Two bilayers divide the system into "an extracellular region" and "a cytosol region". The ionic imbalance between the two regions generates the transmembrane potential. The nanoparticle is inserted 4 nm above the surface of the upper membrane and allows it to interact freely with the membrane during the production run.

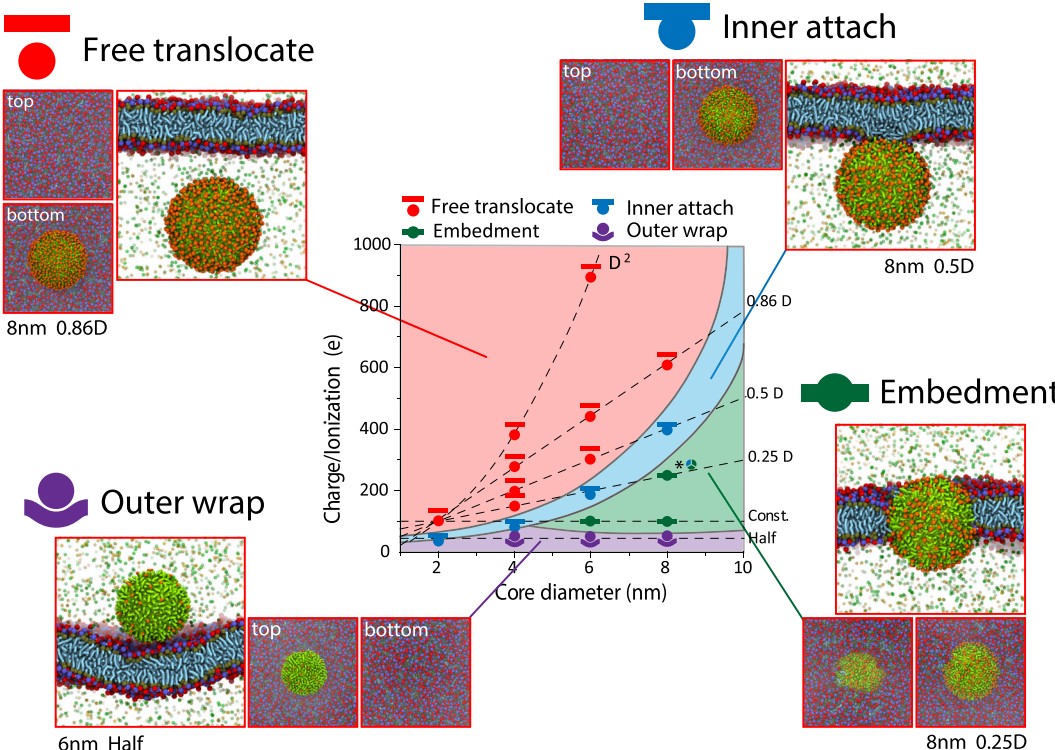

**Fig. 3 Four different types of cellular entry (free translocation, inner attach, embedment, and outer wrap) occur as a result of nanoparticle-cell membrane interactions (for nanoparticles coated with hydrophobic ligands).** Nanoparticles with varying core size and surface charge/ionization are coated with hydrophobic ligands (alkyls). Each point on the graph represents triplicated simulations. *In this triplicate simulations, two are embedment and one is inner attach.

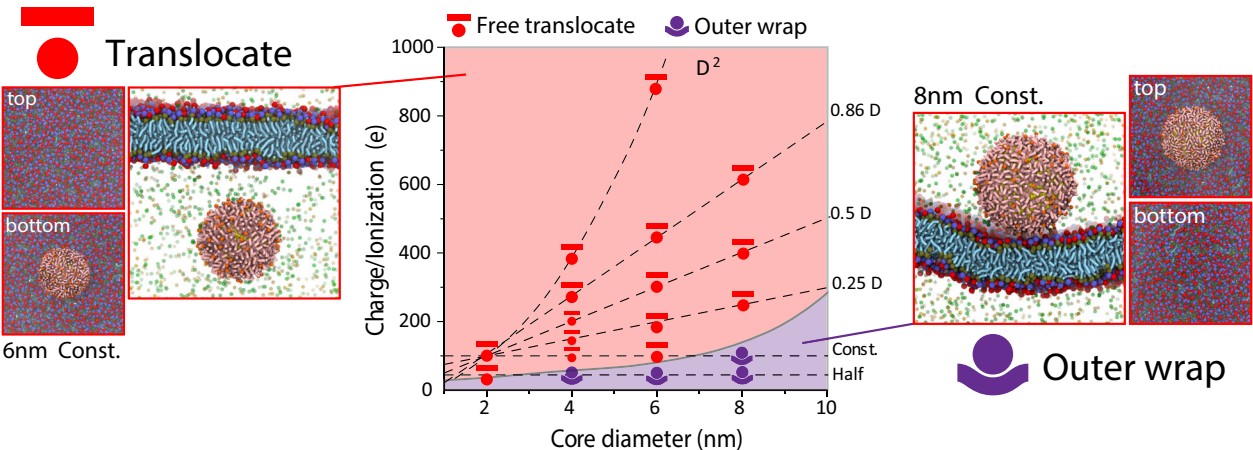

**Fig. 4 Two different types of cellular entry (free translocation and outer wrap) occur as a result of nanoparticle-cell membrane interactions (for nanoparticles coated with hydrophilic ligands).** Nanoparticles with varying core size and surface charge/ionization are coated with hydrophilic ligands (PEG). Each point on the graph has triplicated simulations.

eventually to free translocate (Supplementary Fig. 2). Thus, increasing surface charge/ionization increases the propensity of nanoparticles to bypass the membrane, acting in opposition to the effect of increasing size. Interestingly, however, reducing nanoparticle size seems to be more effective in inducing free translocation than increasing the surface charge (Fig. 2).

Effect of ligand chemistry: Unlike the other two parameters, ligand chemistry is relevant only when the particle is inside the membrane. Shifting ligand chemistry from hydrophobic to hydrophilic will allow the nanoparticle to pass through the membrane instead of embed inside or attach to the surface (Supplementary Fig. 3). It does not have a substantial impact on

the nanoparticles to initiate pore nucleation, although hydrophobic ligands can in some cases insert into the membrane which might facilitate pore nucleation (Supplementary Fig. 4). Overall, increasing ligand hydrophobicity will increase the enthalpic interaction between membrane non-polar interior and the ligands, which helps trap the nanoparticles inside the membrane. Previous atomistic simulation suggests that nanoparticles feature more hydrophobic ligands tend to have a larger free energy gain when inserting into lipid membranes comparing to nanoparticles with less hydrophobic ligands, corrobarting our findings[5].

All three key physicochemical parameters of the nanoparticle need to be considered in designing nanoparticles that would

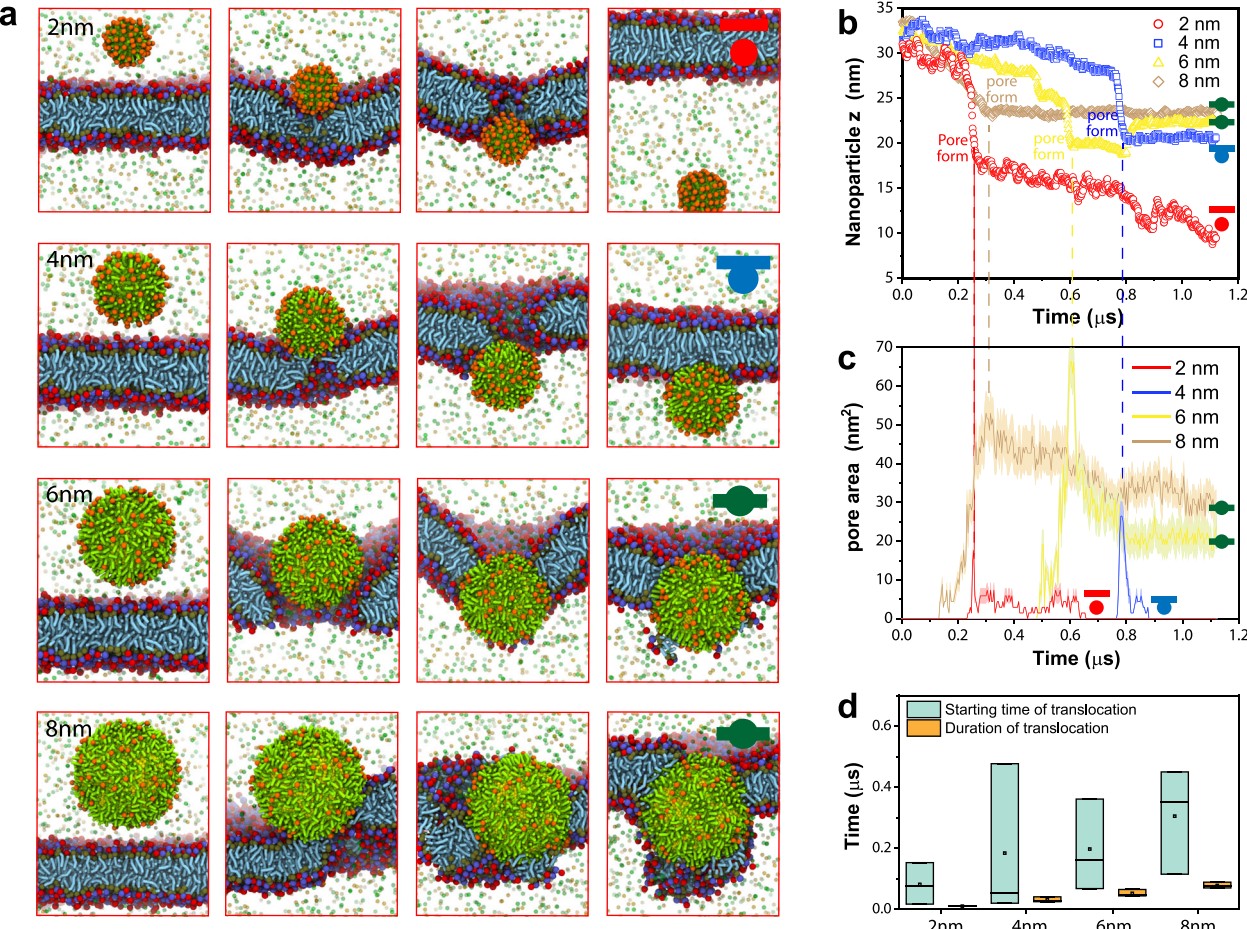

**Fig. 5 Effect of nanoparticle size on translocation. a** Snapshot of translocation process of nanoparticles (100 $e$ and hydrophobic ligand) with varying sizes across model cell membranes for cellular entry. **b** $z$ coordinates of nanoparticles (hydrophobic with surface charges) when interacting with cell membranes. **c** Size of membrane pore vs. time. Gray areas are estimated errors in the area of the pore. **d** Kinetics of NP translocation. Boxes are 25–75% range of data ($n$ = 3, replicated simulations). Black lines inside are median line. Hollowed dots are mean. The duration of translocation is the spanning time of nanoparticle moving through the membrane. For inner attach and embedment, duration time ends when nanoparticle stops moving relative to the membrane.

experience a certain type of translocation. In theory, taking one parameter to the extreme might substantially alleviate the need to finely tune the other two in achieving a certain goal, but experimental constraints, such as the fact that the size of lipid nanoparticles that encapsulate nucleic acids cannot be reduced beyond a certain limit, often make it difficult for this approach to be used in practice.

Nanoparticle shape, degradability, and bonding modes (physisorption and chemisorption) can also affect nanoparticle entry mode. These parameters can somewhat be converted to the three primary parameters (size, surface charge, and ligand hydrophobicity) for estimation. Generally, nanoparticle shape can be converted to size in three different dimensions and their interaction with cell membranes is usually dictated by their largest dimension. Therefore, choosing the largest dimension as the size is a simple way to consider the effect of shape[14]. Although in some cases, the translocation of non-spherical nanoparticles depends on the angle of entry[6]. Nanoparticle degradability usually means the disassociation of coating ligand. By knowing the percentile of remaining ligands on the nanoparticle, one can adjust the surface charge/p$K$a and ligand hydrophobicity accordingly. Bonding modes often render the nanoparticle with a layer of absorbed proteins known as the protein corona. The addition of protein corona can substantially change the physicochemical properites of nanoparticles[30], which has its own field and is not the primary focus here. However, there are

some easy ways to estimate the effect of protein corona here. Protein corona is typically a layer of hydrophobic proteins clustered on the surface of nanoparticles. One can increase size and hydrophobicity to include the effect of the protein corona. Overall, parameters of nanoparticle not investigated here can be roughly converted to the three primary parameters for estimation.

**Membrane curvature.** We observed that before translocation, the nanoparticle attaches to the membrane surface and induces membrane curvature. Then, a pore is generated beneath the nanoparticle, allowing the nanoparticle to quickly pass through the membrane (Fig. 5a, b). The pore-assisted translocation process has been reported previously[14]. Similar to free translocate, pore formation is also observed for inner attach and embedment. For these two outcomes, nanoparticles are unable to completely cross the membrane, but the initial process is the same for all three translocation types (Fig. 5a, b). Pores generated on the membrane are roughly the same size as the nanoparticles (Fig. 5c) and reseal after the translocation of nanoparticles (in embedment, pores reseal as the surrounding lipid close in on the nanoparticle). Interestingly, larger pores appear to reseal faster than smaller pores (4 nm vs 2 nm in Fig. 5c). One explanation is that pores remain open to enable ion influx and efflux which serve to discharge the potential difference across the membrane[14], resulting in the observation that smaller pores remain open longer.

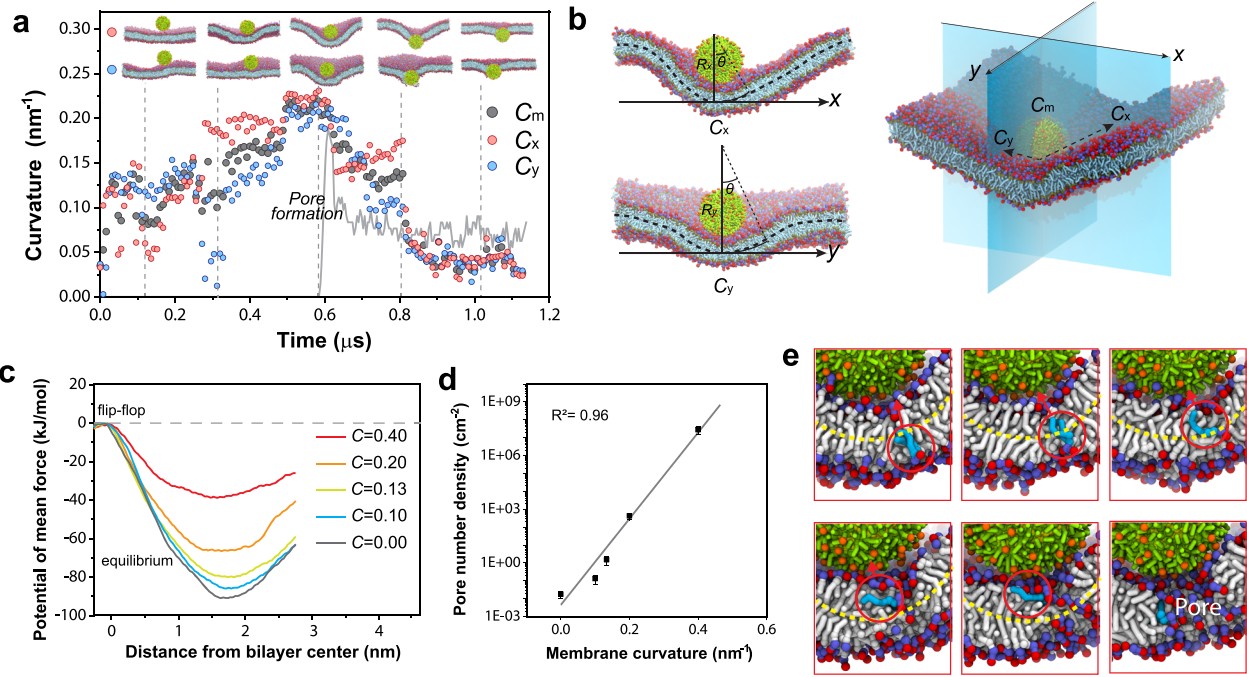

**Fig. 6 Membrane curvature and translocation. a** Membrane curvature (including two principal curvatures and mean curvature) vs. simulation time. Green lines indicate the size of the pore. **b** Illustration of membrane principal curvatures and membrane mean curvature. **c** Potential of mean force (PMF) of lipid flip-flop (from outer leaflet to inner leaflet) in membrane cylinders with curvatures range from 0.00 nm$^{-1}$ (flat membrane) to 0.40 nm$^{-1}$. **d** The estimated probability of pore nucleation density as a function of membrane mean curvature. Error bars are errors from estimating PMF of lipid flip-flop. The gray line is the exponentially fit. **e** Snapshots of initial pore nucleation of the curved membrane in association with nanoparticle translocation (4-nm-0.86R-hydrophobic). A lipid (depicted in blue) flipped from the lower leaflet to the upper leaflet right before pore nucleation (bilayer mid-plane is depicted by the yellow dash line).

The starting time of pore nucleation does not seem to correlate with nanoparticle size and largely varies even amongst triplicated simulations that have the same parameter setup (Fig. 5d), suggesting pore formation is a rather random process. In contrast, the translocation time of the nanoparticle (the time the nanoparticle takes to exit or stop moving after entering the membrane) increases as the nanoparticle becomes larger (Fig. 5d).

Judging from the process of translocation, membrane curvature fluctuates largely and is believed to play a role in the pore nucleation process that triggers subsequent nanoparticle translocation (Fig. 6a). It has been reported that membrane curvature influences lipid tail protrusion behavior[31]. To further understand the effect of membrane curvature on translocation, we characterized the mean curvature of the membrane, $C_m$ (average of the two principal curvatures $C_x$ and $C_y$), at the location beneath the nanoparticle where pore nucleation occurs (Fig. 6b). This vertex point beneath the nanoparticle has the largest $C_m$ in the membrane. We observed that consistent throughout all the simulations, pore nucleation always occurs when $C_m$ reaches its maximum value (Fig. 6a).

It is thus hypothesized that increasing membrane curvature raises the probability of pore nucleation. To test this hypothesis, we estimated the free energy of pore nucleation under different membrane curvatures. The free energy barrier of lipid flip-flop, $\Delta G_{flip}$, can be used to estimate the density of pore nucleation, $\rho$[25]:

$$\rho = \exp\left(-\Delta G_{flip}/kT\right)/A_{lip}$$

where $A_{lip}$ is the area per lipid of the membrane, $k$ is Boltzmann constant, and $T$ is the absolute temperature. Here, we use cylindrical membranes to induce membrane curvatures and the potential of mean force (PMF) of pulling a lipid across the cylindrical membranes with varying curvatures is calculated to obtain $\Delta G_{flip}$. The challenge of sampling the pulling free energy in curved membranes is addressed by adding constraints (details of

implementation can be found in Methods section). It is found that the PMF cost of lipid flip-flop decreases with the increase of the cylindrical membrane's curvature (Fig. 6c). The energy barrier drops abruptly when the radius of the curvature approaches the thickness of the membrane ($C = 0.40$ nm$^{-1}$ or $R = 2.5$ nm), indicating the pores are very likely to be nucleated when the membrane is highly curved. The estimated pore nucleation density is found to increase exponentially with membrane curvature (Fig. 6d). Snapshots that show a lipid flipping from the lower leaflet to the upper leaflet in the curved membrane when pore nucleation occurs under nanoparticle contact (Fig. 6e) corroborates the free energy calculation. Microscopically, curvatures on membrane impose asymmetric stress featuring one leaflet being compressed and the other one stretched. The stretched leaflet possesses high tension which can reduce the free energy cost of pore formation[25]. The breakdown of one leaflet often triggers puncturation on the other leaflet as it is easier to form a water channel with half the non-polar interior[27]. Therefore, high local curvature combined with local electric field triggers pore nucleation beneath the nanoparticles.

## Discussion

Synthesizing the simulation results, we propose a paradigm of the nanoparticle-membranes interface, which features four fundamental elements that affect translocation (Fig. 7). These are physical barriers, chemical barriers, internalization force, and membrane disruption. These are discussed in turn below. The outcome of the translocation type is a result of a combination of these elements. The paradigm can be generalized to a boarder range of scenarios including different nanoparticle and membrane types, and can even be generalized to other types of bio-nano interfaces such as the mucosal interface and the nanoparticle-blood vessels interface.

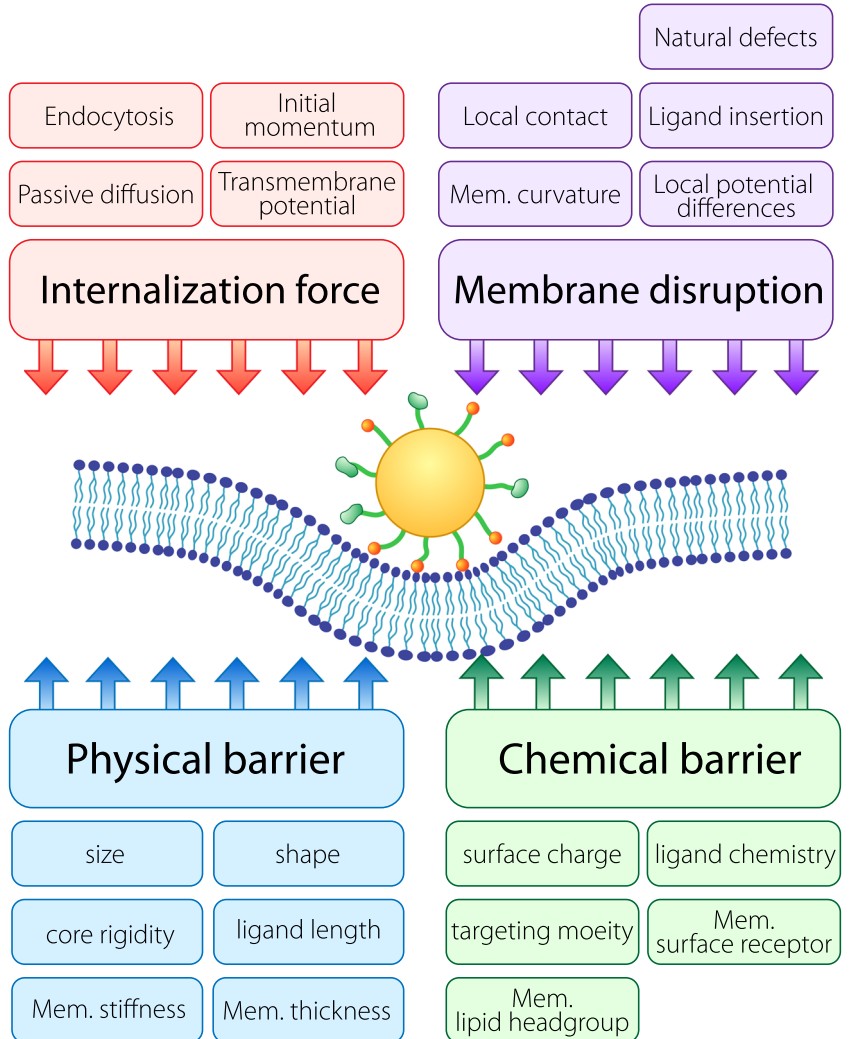

**Fig. 7 Paradigm of the fundamental interactions of nanoparticles at the cell membrane interface.** Four fundamental elements, which are physical barriers, chemical barriers, internalization force, and membrane disruption, regulate the interaction of nanoparticle with cell membrane for translocation. Physical barriers and chemical barriers prevent the nanoparticle from translocating the membrane whereas internalization forces and membrane disruption work in favor of overcoming the membrane. The balance of these forces determines the outcome of translocation.

The physical barrier is determined by both the physical properties of nanoparticles and membranes. It involves considerations including but not limited to nanoparticle size, shape, core rigidity, membrane thickness, and membrane stiffness. For instance, for the translocation process, nanoparticle size and membrane thickness together determine a physical barrier, while for endocytosis, membrane stiffness becomes the major physical barrier due to bending energy associated with membrane vesicle formation.

The chemical barrier plays a large role in governing the chemical interaction between nanoparticles and cell membranes. It involves factors such as nanoparticle surface charge/ionization, ligand chemistry, targeting moiety, and membrane receptors. For small particles such as ions or fullerenes, the chemical barrier of the membrane acts to prevent them from diffusing through or traps them inside[26]. For larger nanoparticles with hydrophobic moieties, the chemical barrier acts to keep particles attached to or embedded in the membrane. Membrane compositions can affect the strength of the chemical barriers. Lipids with longer tails will promote the enthalpic interactions between nanoparticle hydrophobic ligands and the membrane interior which leads to the increase of the barrier strength. Simulation of longer lipid (DOPC) shows that it is slightly harder for nanoparticles to translocate across DOPC membranes than DPPC membranes (Supplementary Fig. 5).

Internalization force is the key driving force for nanoparticle internalization into cells. It can be a result of electrostatic attractions, biological force (e.g. receptor-mediated endocytosis), or other external stimulants. For receptor-mediated endocytosis, the internalization force is determined by the density of receptors on the membrane[29]. For ionized nanoparticles and biomolecules, electrostatic attractions between charged moieties on the nanoparticles and intracellular ions are the driving force. The relative magnitude of the effect of internalization force and that of the physical barrier is crucial in deciding the outcome of translocation. If the internalization force is strong enough, it will certainly allow the nanoparticle to pass through the membrane. For instance, microinjection uses a strong internalization force (initial velocity) to send the nanoparticle inside[32].

Membrane disruption can undermine the strength of membrane barriers and facilitate the process of nanoparticle translocation. Example of modes of membrane disruption includes curvature, membrane softening[33], local potential elevations[27], electroporation, and ligand insertion. Generally, membrane disruption will increase the probability of pore formation on

membranes. Membrane curvature induced by nanoparticle contact is a form of membrane disruption that initiates pore nucleation. An example of a more direct method is electroporation, which generates transient pores on membranes to facilitate the translocation of large molecules[34].

Generally, the internalization force and membrane disruption work in favor of translocation while physical barrier and chemical barrier work against translocation. Design parameters of nanoparticles often affect two or more of these fundamental elements. For example, size/shape can exert an influence on physical barrier but also affects membrane disruption by changing membrane curvature in the contact region; nanoparticle surface charge/p$K$a can affect both chemical barrier and internalization force.

Lastly, a nanoparticle's translocation type has implications for its cellular entry pathway, subsequent intracellular trafficking routes, and biological fate. As is shown in Fig. 8, each translocation type identified herein features a distinct trafficking route and cellular fate. The entry pathway associated with outer wrap is receptor-mediated endocytosis or non-specific endocytosis, both of which lead to endosomal entrapment. Later on, the nanoparticle is either transferred to lysosomal facing degradation or being exocytosed[35,36]. In either route, the nanoparticle is sequestered and cannot access the cytosol. This scenario is the most common one for nanoparticles and presents a substantial barrier for cytosolic delivery. However, nanoparticles with certain designs can escape the sequestration after endocytosis. For instance, lipid nanoparticles functionalized with amines featuring high p$K$a can be ionized in the acidic environment of endosomes and can possibly break out to reach the cytosol[37].

Nanoparticles with free translocation type completely translocate to the other side of the membrane and roam freely in the cytosol. While in the cytosol, nanoparticles can access intracellular machinery or target subcellular organelles. They may remain inside the cell but might be eventually degraded by cytosolic enzymes[32]. Free translocation is a key design criterion for delivery vectors that carry therapeutics to function intracellularly. However, a balancing consideration is that a large number of nanomaterials delivered

inside the cell might cause concern of cytotoxicity due to damage to subcellular organelles and cellular membrane[20].

On the other hand, nanoparticles with the inner attachment translocation type translocate across the membrane and have access to cytosols, but remain bound to the membrane. Some of the attached nanoparticles might come together with endosomes but still attach to the membrane. Eventually, the fate of nanoparticles is membrane localization (no subcellular targeting) with a possibility of enzymatic degradation[38]. Such a configuration of nanoparticles can be used to probe and interact with membrane proteins. Cargos can be released into the cytosol, and nanoparticles are less toxic compared to those that enter through free translocation due to restricted access to subcellular organelles.

Finally, nanoparticles with the embedment translocation type have access to both extracellular fluid and cytosol but are permanently localized on the cell membranes. In some cases, nanoparticles can diffuse laterally on the membrane surface, interacting with other membrane proteins[39]. Such a configuration of nanoparticles can be utilized as transporters on the membrane that allows the passage of certain molecules into and out of the cells. In addition, nanodevices that have the embedment configuration on cell membranes can be used as artificial receptors. The nanoparticles might have an insertion mechanism similar to that of membrane proteins[19].

Since different cellular entry pathways lead to different intracellular routes, one can expect these entry types to trigger different downstream signaling cascades and elicit different cellular responses[4]. The identification and study of these entry types have significant substantial implications for guiding the design of intelligent nanodevices that harvest these cellular responses. It is of particular interest to address the relationship between nanoparticle entry pathways and the types of cellular response they generate, especially for inner attach configurations that were not explored previously.

In summary, we have studied the fundamental interactions at the nanoparticle-cell membrane interface in an effort to understand nanomaterials' cellular entry pathways using computer

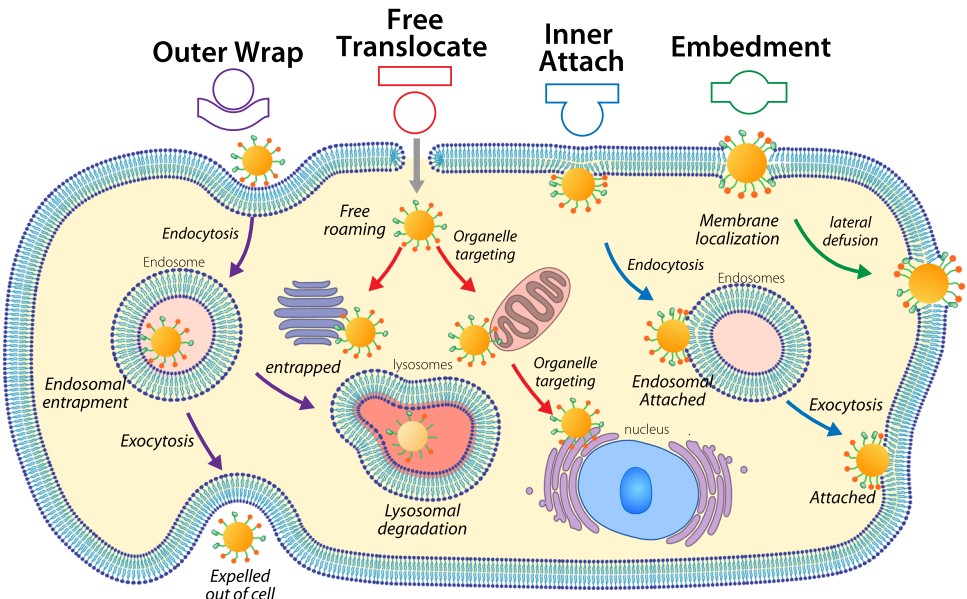

**Fig. 8 Different intracellular trafficking routes and cellular fates of nanoparticles with varying key physicochemical parameters result from the observed four translocation types: outer Wrap, free translocate, inner attach, and embedment.** The fates of nanoparticle include endosomal/lysosomal degradation, exocytosis, intracellular roaming, and membrane localization. Free translocate, inner attach, and embedment allows nanoparticles to access cytosols. The trafficking routes of inner attach and embedment are membrane-bound. Embedment allows the nanoparticle to connect extracellular matrix and cytosols.

simulation. Specifically, we investigated the synergistic effect of size, surface charge/p$K$a, and ligand hydrophobicity of nanoparticle on their interactions with model cell membranes and discovered four types of translocation for cellular entry. We show that the key physicochemical properties of nanoparticles each have a unique role in shaping the translocation outcome, but also have synergistic effects that influence each other. We found that membrane local curvature plays an important role in triggering initial pore nucleation for translocation. The results are further generalized into a paradigm featuring fundamental elements of the nanoparticle-membrane interface that aim to guide the design of nanostructures for specific types of cellular entry pathways. For instance, to achieve free translocate, one can either reduce nanoparticle size or increase their surface charge. To achieve embedment, one needs to include hydrophobic ligands and keep the size and surface charge within a certain zone. It is advised to tune as many parameters as possible at the same time to achieve the goal. Tuning only one parameter typically require a larger range of adjustment and can generate side effects. Furthermore, these findings and the paradigm can be translated to study other bio-nano interfaces (e.g., blood-nano and mucosal-nano interface) for a broader range of applications.

## Methods

**Molecular dynamics and Force fields**. The MARTINI coarse-grained (CG) force field is used for all the simulations[40]. We employ the polarizable water model in the MARTINI framework to give a relatively realistic representation of electrostatic interactions[14]. The time step of the simulation is 0.02 picoseconds. Nanoparticle system normally has the dimensions of $37 \times 37 \times 48$ nm with ~1,450,000 CG atoms. All simulations use periodic boundary conditions. System temperature was controlled at 305 K by velocity rescaling with a time constant of 0.2 ps. Berendsen semi-isotropic pressure coupling was used to control the pressure at 1 atm in lateral and vertical dimensions separately with a time constant of 3.0 ps[41,42]. The cut-off radius of short-range electrostatic interactions is 1.4 nm and the cut-off radius of van de Waals interactions is 1.2 nm. Long-range electrostatics interaction is solved by the Particle Mesh Ewald Method with a mesh density of 0.14 nm[43]. Dielectric constant is set to be 2.5. All simulations were performed using the GROMACS 4.6.7 package[44]. Trajectories were anaylized by VMD 1.9.3.

**Nanoparticles**. The structure of monolayer-protected nanoparticle adopted a motif we developed previously under the MARTINI coarse-grained framework[8]. The coordination of CG atoms is mapped based on an atomistic model. Ligands are attached to the core through Au-S bonds. The coordination of the core-ligand bond is obtained by simulated annealing of the atomistic model. Generally, the ligand coating density reduces as the size of the nanoparticle increases (0.86D regimes) due to steric interactions between ligands (Supplementary Table 1). For nanoparticles in the $D^2$ regimes, the ligand density is artificially kept constant to ensure the scaling of surface charge based on the surface area. The charge/ionization is randomly assigned to the terminals of available ligands on the metal core. Ligands are either alkyl-thiol or polyethylene glycol (PEG)-thiol. The parameters of PEG under MARTINI framework is adapted from previous work[45]. The hydrodynamic diameter of nanoparticles is estimated based on their radius of gyration (Supplementary Fig. 1).

**Model cell membranes**. The model membrane system has is composed patches of (Dipalmitoylphosphaticlylcholine) DPPC bilayer, each containing 4128 lipids. With periodic boundary conditions, the system is divided into two separate regions (Fig. 2b). The top region is the "extracellular region" while the bottom region is the "cytosolic region". The ionic imbalance is implemented between these two regions to create a transmembrane electric potential across the bilayer[14], emulating the potential exists across animal plasma membranes. The transmembrane potential scales linearly with the ionic imbalance added between the two regions[46]. Here, the model membrane system in this manuscript has a 120 ionic imbalance, which results in ~−1.5 V potential in the cytosolic region. The system has ~446,000 polarizable water in the two regions Physiological concentration (150 mM) of NaCl was introduced to the polarizable water, resulting in ~4000 sodium and chloride ions. The model cell membrane system was simulated for 100 ns to equilibrate.

**Nanoparticle-cell membrane systems**. All nanoparticles are placed in the extracellular region with their bottom 3 nm above the membrane. After the insertion of the charged/ionized nanoparticles, counterions are added to the solvent to neutralize the system. After energy minimization, the system was simulated with the nanoparticle core and the membranes constrained by a harmonic potential for 40 ns to equilibrate ligands, solvent, and counterions. Then the system was simulated for 1.2 μs for

production runs with the constraint released. Each case in the three-dimensional parameter space is simulated independently three times. The triplicated simulations have identical setup but have randomly generated initial velocities.

**Membrane analysis**. Membrane pore area is calculated by a discretization method. The membrane is meshed by a $30 \times 30$ grid into columns with a size of 1.23 nm × 1.23 nm. Each column that does not include any phosphate group is counted as part of the pore. Total pore areas are the sum of the area of the columns counted.

Membrane curvature is calculated by a similar discretization method using reference points that help define the projected curved surface (Supplementary Fig. 6). Likewise, the membrane is divided by a $30 \times 30$ grid. At each grid, the reference point of the curved surface is determined as the averaged positions of the phosphate group in the bottom leaflet within a radius of 3 nm. Then, reference normal, $\mathbf{Z_i}$, that is perpendicular to the bilayer surface at each reference point is calculated using average positions of the phosphate group and the tail carbons. $\mathbf{Z_i}$ and its adjacent reference normal, $\mathbf{Z_{i+1}}$, are used to determine the curvature angle $\theta_i$ and the reference distance $D_i$. The radius of the curvature can be then calculated using $R_i = D_i/\tan(\theta_i)$. The curvature at each reference point, $C_i$, is the reciprocal of the radius $R_i$. The calculation is done in both $x$ and $y$ plane separately, yielding two principle curvatures $C_x$ and $C_y$. The mean curvature is the average of the two principal curvatures. In the area of the pore, the lipid that is out of its regular conformation is discarded when calculating the reference points and the reference normal.

**Free energy of pore nucleation on curved membranes**. Curved membranes cylinders are constructed using BUMPy (Supplementary Fig. 7)[47]. The 10-nm long DPPC cylinders with radii of 2.5, 5, 7.5, and 10 nm (resulting in membrane curvatures of 0.4, 0.2, 0.13, and 0.1 nm$^{-1}$) were built respectively (Supplementary Table 2). To calculate the free energy of lipid flip-flop in the curved cylinder, the potential of mean force (PMF) along a pulling coordinate which is perpendicular to the membrane surface and toward the center of the cylinders was calculated[25]. To prevent the cylinders from deforming during the pulling, the inner leaflet of the membrane is constrained by a harmonic force (Supplementary Fig. 8). Umbrella sampling is used to calculate the PMF along the pulling coordinate[48]. The space between each sampling window is 0.1 nm in a total sampling distance of 5 nm. The restraint force for the pulling has a force constant of 5000 kJ mol$^{-1}$ nm$^{-2}$. For each sampling simulations, the system is equilibrated for 20 ns with a 100-ns production run. The free energy curve is obtained by the Weighted Histogram Analysis Method[49].

**Statistics and reproducibility**. For each combination of simulation in the three-dimensional parameter space, there are three replicates. Each replicate has the identical setup expect the initial velocity following Boltzmann distribution were scrambled. Simulation of the typical system in the manuscript (~1.5 M atoms) were run on a 64-core AMD node in parallel mode with 64 threads. Data are collected from 1.2 μs trajectory files with an interval of 0.4 ns between frames. The Weighted Histogram Analysis method was performed within GROMACS 4.6.7 package. Probablity of pore density vs membrane curvature (five sampling points) were fitted to a single exponential growth (OriginPro).

**Reporting summary**. Further information on research design is available in the Nature Research Reporting Summary linked to this article.

## Data availability

All data are available from the corresponding author upon reasonable request.

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

## Acknowledgements

This work was supported by the NSF through the NSF MRSEC program at MIT under grant No. DMR-0819762. Computation resources from Massachusetts Green High-Performance Computing Center (MGHPCC) are gratefully appreciated. The authors acknowledge the support from Dr. Giovanni Traverso and Dr. Robert Langer at Koch Institute of Integrative Cancer Research at MIT.

## Author contributions

J.L. and A.A. designed the study, analyzed the data, and proposed the paradigm of the nanoparticle-membrane interface. J.L. and G.Z. performed the simulations. C.L. and R.D. input valuable ideas that helped conceive the paradigm. J.L., L.M., and A.A. wrote the manuscript. A.A. supervised the study. All authors discussed the results and assisted in the preparation of the manuscript.

## Competing interests

The authors declare no competing interests.
