## [Peer Review File · Communications Biology]

Reviewers' comments:

Reviewer #1 (Remarks to the Author):

Review for Manuscript#: COMMSBIO-19-1005

The manuscript titled "Understanding Fundamental Physicochemical Properties of Nanoparticles at the Cell Membrane Interface: From Translocation Type to Intracellular Trafficking" by Jiaqi Lin et al., is very interesting from computational investigation point of view for nanomaterials and their mode of entry in the cell membrane. The authors have provided the in depth mechanism of action with subtle proof which was not answered by previous findings. Overall, the manuscript looks novel with respect to nanomaterials design aspects and their mode of entry and interaction at cell membrane. However, I have few queries/concern which should be addressed satisfactorily by authors.

Comments:

- 1) The role of physiochemical properties of nanomaterials such as size, charge/pKa and ligand chemistry has been investigated in the present work. What about shape, degradability and bonding modes (physisorption and chemisorption) of nanoparticle? Will it affect entry and translocation by cell membrane? If no, then why?. Authors need to comment/investigate these aspects.
- 2) Line no. 121-123. Authors focused on nanoparticles with size ranging from 3 nm to 15 nm, mentioning that they are on scale of proteins in the body that tend to have complex interactions with the cell membranes. Do these correlations have any evidence? If yes, kindly include the references or include any comment from previous reports.
- 3) At what pH, all the simulation studies have been carried out?
- 4) Give reference at line no. 151 which mentions that DPPC represents the phospholipids in the living cell membranes.
- 5) Line 200-203 mentions that "For instance, nanoparticles with hydrophobic ligand and 100 e surface charge/ionization, as nanoparticle size increases, the translocation type changes from Free translocate to Inner attach and then to Embedment". Do the size of nanoparticle affects outer wrap translocation?
- 6) Line 220-222- "Overall, increasing ligand hydrophobicity will increase the enthalpic interaction between membrane non-polar interior and the ligands, which helps trap the nanoparticles inside the membrane" Can authors correlate this statement/finding/conclusion with any other previous reported study or fact?
- 7) Authors are requested to mention few design parameters for nanoparticles which have been concluded from this study in conclusion section. Few highlights on do's and don'ts while designing the nanoparticles will be key highlight from this study which can help to other researchers while designing the particles.
- 8) In the methods section, authors mentioned the Dielectric constant value of 2.5. What is the basis of selecting this value? As, the constant can range from 2-16 for such computational study which affects the overall charge of water/environment.
- 9) Line 467: Figure S?. Please correct this.

10) What is the basis of selecting DPPC model only? There are many other CG models available for membrane. Can author show a simulation study using same nanoparticle and other membrane model? What would be the conclusion in such case?

Reviewer #2 (Remarks to the Author):

The manuscript by Lin et al. investigated the interactions between monolayer-protected nanoparticles (NPs) and model cell membranes using coarse-grained molecular dynamics simulations. Their results showed that there were four different types of nanoparticle translocation, i.e., outer wrapping, free translocation, inner attach, and embedment. These different translocation types greatly depended on the NP size, surface charge, and ligand chemistry. Moreover, the authors also gave a useful discussion on the implication of nanoparticle translocation type on its cellular entry pathway, subsequent intracellular trafficking routes, and biological fate.

Overall, I would appreciate the motivation of this computational work since recently more and more attention has been paid to the nano-bio interaction at the interface (especially the nanoparticle-cell membrane interaction). Besides, the authors systematically studied the NP-membrane interaction and observed four translocation modes, and more importantly revealed the underlying mechanism of the relationship between the modes and NP physicochemical properties, which I think should provide useful guidelines on the design of nanomaterials. On the basis of above reasons, I would recommend its publication on *Communications Biology* after the following comments are well responded.

1. The literature references here are far from appropriate. The authors missed a lot of literatures closely related to the topic discussed here. For example, the pioneering study on the role of physicochemical property of ligands on the cellular uptake [*Biomaterials*, 33, 5798-5802 (2012)], the first study on spontaneous penetration (free translocation) of nanoparticle through cell membranes [*ACS Nano* 6, 1230-1238 (2012)], and some latest reviews [*Adv. Colloid Interfac.* 218, 48-68 (2015)], [*Small* 11, 1055-1071 (2015)].
2. In my opinion, PEG or alkyls chains do not carry charges. How did the author deal with the charges on the NP? Is there any experimental evidence for the choice here?
3. The author claimed that "inner attach and embedment configurations were not explored previously". To my knowledge, the inner attach may not be explored previously, but the embedment has been largely explored, the authors may refer to the reviews as I mentioned above.
4. Not all supplementary figures occurred in the main text, and their sequence was not in order. And on page 15, line 467, Figure S?

Referee expertise:

Referee #1: Bioinformatics, Structural Biology, Biophysics and Molecular Dynamics Simulations

Referee #2: Nanoparticle-cell membrane interaction, MD simulations

Reviewers' comments:

Reviewer #1 (Remarks to the Author):

Review for Manuscript#: COMMSBIO-19-1005

The manuscript titled "Understanding Fundamental Physicochemical Properties of Nanoparticles at the Cell Membrane Interface: From Translocation Type to Intracellular Trafficking" by Jiaqi Lin et al., is very interesting from computational investigation point of view for nanomaterials and their mode of entry in the cell membrane. The authors have provided the in depth mechanism of action with subtle proof which was not answered by previous findings. Overall, the manuscript looks novel with respect to nanomaterials design aspects and their mode of entry and interaction at cell membrane. However, I have few queries/concern which should be addressed satisfactorily by authors.

Comments:

1) The role of physicochemical properties of nanomaterials such as size, charge/pKa and ligand chemistry has been investigated in the present work. What about shape, degradability and bonding modes (physisorption and chemisorption) of nanoparticle? Will it affect entry and translocation by cell membrane? If no, then why?. Authors need to comment/investigate these aspects.

Yes. Shape, degradability, and bonding modes will definitely affect nanoparticle entry mode. These parameters can somewhat be converted to the three primary parameters (size, surface charge, ligand hydrophobicity) for estimation. Generally, nanoparticle shape can be converted to size in three different dimensions and their interaction with cell membranes is usually dictated by their largest dimension. Therefore, choosing the largest dimension as the size is a simple way to consider the effect of shape (ACS Nano, 2013, 7 10799-808). Although in some cases, the translocation of non-spherical nanoparticles depends on the angle of entry (Nature Nanotechnology, 2010, 5, 579-83). Nanoparticle degradability usually means the disassociation of coating ligand. By knowing the percentile of remaining ligands on the nanoparticle, one can adjust the surface charge/pKa and ligand hydrophobicity accordingly. Bonding modes often render the nanoparticle with a layer of absorbed proteins known as the protein corona. Addition of protein corona can significantly change the physicochemical properties of nanoparticles (Chemical Society Reviews, 2012, 41, 2780-2799), which is not the primary focus of the manuscript. However, there are some easy ways to estimate the effect of protein corona here. Protein corona is typically a layer of hydrophobic proteins clustered on the surface of

nanoparticles. One can increase size and hydrophobicity to include the effect of the protein corona. Overall, parameters of nanoparticle not investigated here can be roughly converted to the three primary parameters for estimation.

The above discussion is added to the manuscript on Page 8.

2) Line no. 121-123. Authors focused on nanoparticles with size ranging from 3 nm to 15 nm, mentioning that they are on scale of proteins in the body that tend to have complex interactions with the cell membranes. Do these correlations have any evidence? If yes, kindly include the references or include any comment from previous reports.

Smaller size objects (<1 nm) such as ions and water molecules enter cell membranes via permeation (Chemistry and Physics of Lipids, 1996, 40, 167-188). Larger size objects enter cell typically through endocytosis, which includes phagocytosis, pinocytosis, receptor-mediated endocytosis and other non-specific endocytosis (Nature Reviews Molecular Cell Biology, 2007, 8, 603-612). In the middle size range that is comparable to the thickness of cell membranes lies the boundary between permeation (direction translocation) and endocytosis. Protein is in such a size range (from 3 nm to 20 nm) and demonstrates complex interactions with cell membranes (Molecules, 2017, 22, 26-28). Nanoparticles within such size range (~10 nm) show interesting interactions with cell membranes (Nano Letters, 2010, 10, 2543). These references and some of the comments are included in the manuscript.

3) At what pH, all the simulation studies have been carried out?

The simulations are carried out at neutral pH. Molecular dynamics simulations normally are not able to reflect the change of pH. The simulations employ polarizable water, which gives better electrostatic interactions. However, it still does not allow the disassociation of protons (PLoS computational biology, 2010, 6, e1000810). Here we predetermine the charge of active moieties using their pKa and assuming a neutral pH environment. For instance, ammonium ion with a pKa value around ~10.5 features a positive charge at neutral pH. This has been further clarified in the manuscript.

4) Give reference at line no. 151 which mentions that DPPC represents the phospholipids in the living cell membranes.

Several previous simulation studies use DPPC to represent cell membranes: Nature Nanotechnology, 2008, 3, 262-368; JACS, 2006, 128, 12462-7. These references are added to the manuscript.

5) Line 200-203 mentions that "For instance, nanoparticles with hydrophobic ligand and 100 e surface charge/ionization, as nanoparticle size increases, the translocation type changes from Free translocate to Inner attach and then to Embedment". Do the size of nanoparticle affects outer wrap translocation?

Yes. In the case of hydrophilic nanoparticle with 100 e surface charge, increasing size change translocation type from Free translocate to Outer wrap. Within the realm of Outer wrap, the size of nanoparticles also affects the rate of translocation. For non-specific endocytosis, larger size nanoparticle is more likely to be internalized due to low curvature induced on the membrane but the entire process might take a longer time. For receptor-mediated endocytosis, nanoparticle

size and receptor density together regulate the rate of uptake, which has been investigated by Zhang et al. (Advanced Materials, 2009, 21, 419-424; Physical Review Letters, 2010, 105, 1-4). The above comments are partly added to the manuscript.

6) Line 220-222- “Overall, increasing ligand hydrophobicity will increase the enthalpic interaction between membrane non-polar interior and the ligands, which helps trap the nanoparticles inside the membrane” Can authors correlate this statement/finding/conclusion with any other previous reported study or fact?

This is one of the interesting findings of the manuscript. One previous study shows that the free energy of nanoparticle insertion into lipid membranes decrease as nanoparticle feature more hydrophobic ligands (Nano Letters, 2013, 13, 4060-4067). Additionally, it has been reported that hydrophobic peptides are able to generate membrane pores and stabilize pores by staying at the inner rim of the pores (Biochimica et Biophysica Acta – Biomembranes, 2010, 1798, 1494-1502), which correlates the enthalpic interaction between the hydrophobic moieties and the membrane interior observed in our simulation. These studies are added in the manuscript

7) Authors are requested to mention few design parameters for nanoparticles which have been concluded from this study in conclusion section. Few highlights on do's and don'ts while designing the nanoparticles will be key highlight from this study which can help to other researchers while designing the particles.

We thank the reviewer for this suggestion. Some design guidelines have been added to the conclusion.

8) In the methods section, authors mentioned the Dielectric constant value of 2.5. What is the basis of selecting this value? As, the constant can range from 2-16 for such computational study which affects the overall charge of water/environment.

The value is based on the parameterization of the MARTINI polarizable water model (Ref. 39 PLoS computational biology, 2010, 6, e1000810), which gives the right electrostatic interactions between polarizable waters and other charged groups under the MARTINI framework.

9) Line 467: Figure S?. Please correct this.

We apologize for this typo. It has been corrected (as Figure S1).

10) What is the basis of selecting DPPC model only? There are many other CG models available for membrane. Can author show a simulation study using same nanoparticle and other membrane model? What would be the conclusion in such case?

This is a very good question. We have performed additional simulations using a DOPC membrane and different sized hydrophobic nanoparticles (from 2 nm to 8 nm) with 100 e surface charge. The result shows that it is harder for nanoparticles to translocate across DOPC membranes than DPPC membranes. Specifically, the translocation type of 4 nm nanoparticle change from Inner Attach to Embedment with DOPC membranes (Figure S8). This is probably

due to the longer tail length of DOPCis which increased the enthalpic interactions between membrane interior and the hydrophobic ligands on the nanoparticle. Therefore, the composition of lipids can also affect the translocation types for a given physicochemical property of nanoparticles. The above comments have been added to the manuscript and the results have been included in supplementary materials (Figure S8).

Reviewer #2 (Remarks to the Author):

The manuscript by Lin et al. investigated the interactions between monolayer-protected nanoparticles (NPs) and model cell membranes using coarse-grained molecular dynamics simulations. Their results showed that there were four different types of nanoparticle translocation, i.e., outer wrapping, free translocation, inner attach, and embedment. These different translocation types greatly depended on the NP size, surface charge, and ligand chemistry. Moreover, the authors also gave a useful discussion on the implication of nanoparticle translocation type on its cellular entry pathway, subsequent intracellular trafficking routes, and biological fate.

Overall, I would appreciate the motivation of this computational work since recently more and more attention has been paid to the nano-bio interaction at the interface (especially the nanoparticle-cell membrane interaction). Besides, the authors systematically studied the NP-membrane interaction and observed four translocation modes, and more importantly revealed the underlying mechanism of the relationship between the modes and NP physicochemical properties, which I think should provide useful guidelines on the design of nanomaterials. On the basis of above reasons, I would recommend its publication on Communications Biology after the following comments are well responded.

1. The literature references here are far from appropriate. The authors missed a lot of literatures closely related to the topic discussed here. For example, the pioneering study on the role of physicochemical property of ligands on the cellular uptake [Biomaterials, 33, 5798-5802 (2012)], the first study on spontaneous penetration (free translocation) of nanoparticle through cell membranes [ACS Nano 6, 1230-1238 (2012)], and some latest reviews [Adv. Colloid Interfac. 218, 48-68 (2015)], [Small 11, 1055-1071 (2015)].

We apologize for the absence of some of the important relevant references. The references mentioned here are all added to the manuscript. In addition, we have added a dozen new references by answering the reviewers' questions.

2. In my opinion, PEG or alkyls chains do not carry charges. How did the author deal with the charges on the NP? Is there any experimental evidence for the choice here?

PEG or alkyls chains themselves do not carry charges. However, they are terminated with amine groups that carry positive charges in a neutral pH environment. The charges are at the end of the chains (shown as orange beads in Figure 1). There are many experimental investigations on charged nanoparticles that feature this motif (Nature materials, 2008, 7, 588-595; Accounts of chemical research, 2013, 46, 681-691). The references have been added to the manuscript.

3. The author claimed that “inner attach and embedment configurations were not explored previously”. To my knowledge, the inner attach may not be explored previously, but the embedment has been largely explored, the authors may refer to the reviews as I mentioned above.

We apologize for the overlook on the matter. The above comment has been revised to include only the Inner Attach configuration.

4. Not all supplementary figures occurred in the main text, and their sequence was not in order. And on page 15, line 467, Figure S?

We apologize for the overlook of the sequence and the typo. All the issues mentioned above have been addressed.